# Investigation of electromagnetic wave propagation characteristics across various frequencies in porous media of goaf

Qingsong Zhang[1,2], Weikang Zhao[1,2], Hui Zhuo[1,2]*, Long Lin[1,2], Zuojian Li[1,2]

1 College of Safety Science and Engineering, Anhui University of Science and Technology, Huainan, Anhui, China, 2 Engineering Technology Research Centre for Safe and Efficient Coal Mining, Anhui University of Science and Technology, Huainan, Anhui, China

* zhuohui1130@126.com

**Data Availability Statement:** All relevant data are within the manuscript and its Supporting Information files.

## Abstract

The reliable long-distance transmission of electromagnetic wave signals within goaf is fundamental for the implementation of wireless monitoring and early warning systems for goaf-related disasters. This paper establishes an experimental platform for electromagnetic wave signal transmission within goaf and develops a propagation model for electromagnetic waves in the porous media of goaf. The transmission characteristics of electromagnetic waves at various frequencies within the porous media environment of goaf are investigated through experimental and numerical simulation approaches. The results indicate that the received signal intensity of electromagnetic waves across different frequency bands diminishes with increasing propagation distance in the lossy environment of the goaf. Initially, the decay follows a logarithmic pattern, whereas, at later stages, the attenuation exhibits a gradual and smooth decrease. As the frequency increases, the initial attenuation amplitude of electromagnetic wave intensity rises; however, subsequent attenuation is largely unaffected by frequency, with the later attenuation rate being proportional to porosity. Electromagnetic waves at a frequency of 700 MHz exhibit a low attenuation coefficient under both experimental and simulated conditions, demonstrating superior stability and reliability. This frequency significantly enhances the overall performance of the communication system and is suitable for use as the operational frequency band in wireless sensor networks.

## 1. Introduction

Spontaneous combustion of coal in goaf poses a critical threat to both mine safety and operational productivity. This phenomenon not only produces toxic and hazardous gases, such as $C_xH_y$ and CO, but also precipitates secondary catastrophes, including gas explosions, which result in severe casualties and extensive property damage [1–4]. xsConsequently, developing an effective monitoring and early warning system for coal spontaneous combustion is essential for preventing and managing mine fires. Currently, wired and fixed sensor monitoring networks are extensively utilized in goaf; however, they encounter issues such as challenging

**Funding:** National Natural Science Foundation of China (52204192), Anhui Provincial Key Research and Development Project (2022m07020006). The funder defines the research direction, designs the research plan, supervises project progress, and ensures that research and experiments proceed according to plan.

wiring, high maintenance costs, and limited environmental adaptability [5,6]. In line with the new paradigm of precise identification, monitoring, and early warning of dynamic disaster risks in coal mines, as well as the concept of 'smart mines' [7–9]the technology for monitoring and early warning of goaf disasters increasingly integrates wireless ad hoc network technology with temperature sensors. This technology has transformed the traditional wired approach to goaf disaster monitoring, enhancing both the accuracy and timeliness of monitoring and early warning systems [10,11], while effectively addressing the limitations of existing fixed sensor networks. However, issues such as signal attenuation and electromagnetic interference within the goaf result in unstable signal quality and significantly constrain the wireless transmission range.

The propagation characteristics of electromagnetic waves at different frequencies in porous media are a focal point of research, as the frequency of these waves significantly influences their propagation properties. Low-frequency electromagnetic waves possess a greater penetration capability, enabling them to traverse thicker coal seams and rock layers; however, they are susceptible to interference from geological scattering and multipath effects during propagation. Although high-frequency electromagnetic waves have relatively weaker penetration capabilities, their shorter wavelengths make them suitable for high-resolution sensing and short-range communication. This frequency dependence provides a basis for optimizing the design of wireless communication systems across various application scenarios [12,13]. The complex geological structures of goaf regions can induce significant multipath effects, complicating signal propagation, with distinct multipath interference phenomena observed at different frequencies [14]. To gain a deeper understanding of the characteristics of electromagnetic wave propagation in goaf regions, numerical simulation techniques are extensively employed. Recent studies have utilized the Finite-Difference Time-Domain (FDTD) method to simulate electromagnetic wave propagation paths and signal superposition in complex environments. The FDTD method has demonstrated high accuracy in modeling the interactions between electromagnetic waves and complex geological structures, allowing for detailed characterization of reflection, attenuation, and scattering features [15]. Ray tracing methods have been widely employed in recent years to simulate the propagation paths of electromagnetic waves in complex goaf environments, particularly in scenarios where multipath effects are pronounced. Research indicates that ray tracing methods, when integrated with actual geological conditions, can more effectively predict the attenuation characteristics of signals and multipath interference phenomena [16].By introducing various numerical simulation techniques and frequency optimization strategies, researchers can more accurately predict the propagation paths and attenuation patterns of electromagnetic waves in goaf regions, further enhancing detection capabilities in complex mining environments.

Currently, wireless monitoring systems for goaf have demonstrated preliminary advancements in both technological application and system development. However, a critical factor for these systems is the propagation distance of wireless electromagnetic waves within the goaf. The goaf is characterized by complex terrain, numerous obstacles, and significant spatial limitations. Within this context, electromagnetic wave propagation experiences rapid attenuation due to environmental interference. Consequently, the effective transmission distance of information via wireless communication is a critical determinant of the system's practicality. Should the spacing of wireless communication be inadequate, additional monitoring equipment will be required to comprehensively cover the goaf, thereby escalating the costs associated with the monitoring system layout and complicating system maintenance and management. The long-distance propagation of wireless signals within the goaf not only ensures extensive communication coverage but also significantly enhances the system's overall efficiency. Consequently, this study develops a simulation platform to accurately replicate the

porous media environment of the goaf. It establishes an electromagnetic wave propagation model for these media, conducts both experimental and numerical simulations, and investigates the propagation loss, signal penetration capability, and frequency dependence of electromagnetic waves at various frequencies within the goaf environment. This work aims to provide a theoretical foundation for advancing wireless monitoring systems in goaf and enhancing the transparency of goaf environments.

## 2 Construction of goaf simulation platform

### 2.1 Characteristics of the wireless communication environment and signal frequency selection in goafs

As the working face progresses, the overlying coal and rock masses gradually collapse and become compacted. According to the 'O' ring theory, the compactness of the collapsed coal and rock masses within different regions of the goaf exhibits non-uniformity [17]. In the natural accumulation zones of the caving coal and rock mass adjacent to the intake air, return air, and working face, the porosity is relatively high. Conversely, in areas subject to significant overburden pressure within the goaf, the caving coal and rock mass undergoes gradual compaction, resulting in reduced porosity. The intricate pore structure of the goaf results in a complex and variable propagation path for electromagnetic wave signals [18]. This variability gives rise to multiple transmission modes, including direct, refraction, reflection, and transmission, which contribute to the instability of signal interference and data transmission, thereby impacting the quality of signal reception in the communication system. Furthermore, as electromagnetic waves propagate from the depths of the goaf to the external environment, they become increasingly influenced by the complex subterranean electromagnetic environment. This effect is especially pronounced with greater quantities of equipment at the working face, where electromagnetic interference becomes particularly significant [19–22].The transmission path of electromagnetic waves within the coal-rock medium is illustrated in Fig 1.

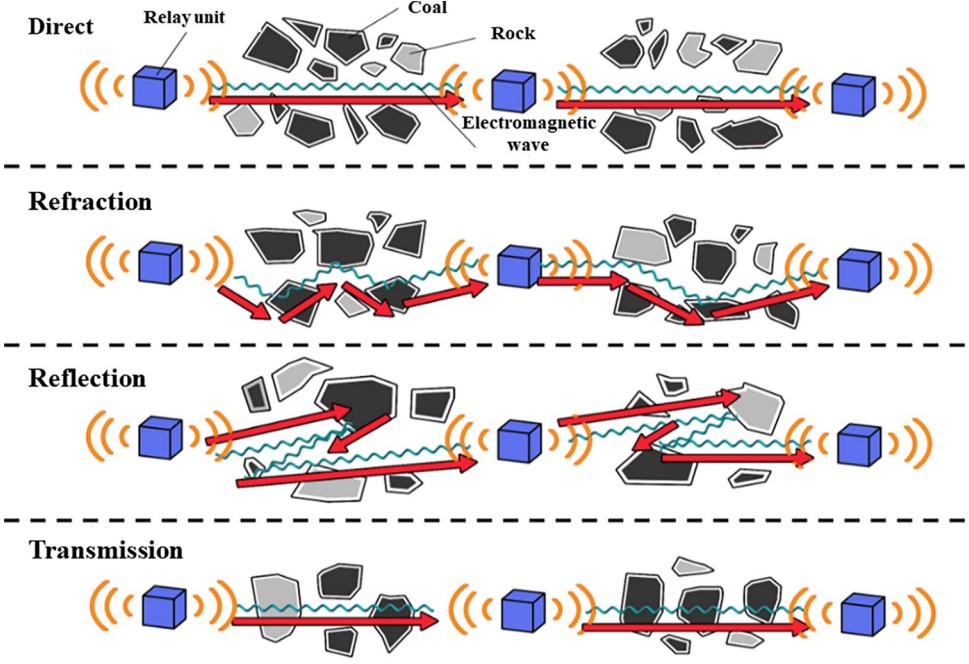

**Fig 1. EM wave transmission in coal-rock media.**

Lower frequency electromagnetic waves exhibit enhanced resistance to attenuation within the transmission medium which translates to a greater penetration capability of the waves [23–25]. The UHF (300 MHz to 3 GHz) band is characterized by its long communication range and robust penetration capabilities, making it a primary focus of research in mobile communication wave propagation [26–28]. Spectrum planning in numerous countries and regions designates frequency bands above 600 MHz for communication services, thereby ensuring the validity and effectiveness of signal propagation experiments. Signals in the 600 to 900 MHz frequency range exhibit long wavelengths, strong penetration capabilities, and low propagation loss, ensuring extended communication distances and sustained signal strength in the goaf environment. Furthermore, the 600 to 900 MHz band offers moderate bandwidth, which supports adequate data transmission rates within the goaf environment. Therefore, this study thoroughly evaluates the stability and reliability of signal transmission within the goaf environment using four frequencies: 600, 700, 800, and 900 MHz.

## 2.2 Experimental platform and measuring equipment

The experimental platform primarily consists of a metal pipe structure designed to simulate the electromagnetic wave propagation environment within a goaf. The specific construction process involves: Based on the selected frequency range of 600–900 MHz, and applying wave theory and transmission line theory [29,30], the cutoff frequency of the pipeline can be calculated using the formula:

$$\lambda = \frac{C}{f_c} \tag{1}$$

For a circular waveguide, $f_c$ = 600 MHz, C≈3×10$^8$, the relationship between the cutoff wavelength and the diameter is as follows:

$$\lambda = \frac{2.405 \times D}{1} \tag{2}$$

Calculated according to the equation:

$$D \approx 0.208m$$

According to calculations, the diameter of a circular pipeline with a cutoff frequency below 600 MHz is approximately 0.208 meters. To facilitate the experiment, the pipeline diameter was increased to 0.75 meters. This adjustment not only provides additional tolerance but also helps avoid potential issues arising from non-compliance with design standards. Therefore, the main structure of the platform was designed as a metal steel pipe with an inner diameter of 0.75 m and a length of 5 m. The pipeline is filled with a mixed medium of granite and quartzite to simulate the goaf environment., with measurements recorded at 1-meter intervals. The porosity is controlled for each meter along the length of the simulation platform, thereby simulating the effects of varying pore sizes and rock types in a real goaf on electromagnetic wave propagation. Electromagnetic signal transmitters and spectrum analyzers are positioned at both ends of the pipeline. The transmitting antenna is connected to a signal generator to emit electromagnetic wave signals across various frequency bands. The receiving antenna captures the transmitted signal and measures its attenuation and scattering characteristics. Simulation platform for porous media environment in goaf is illustrated in Fig 2. Schematic diagram of the interior of the goaf simulation platform is illustrated in Fig 3.

Ensuring the accuracy of the measured data during the experiment is critically dependent on the calibration of the spectrum analyzer and signal generator. Prior to the formal

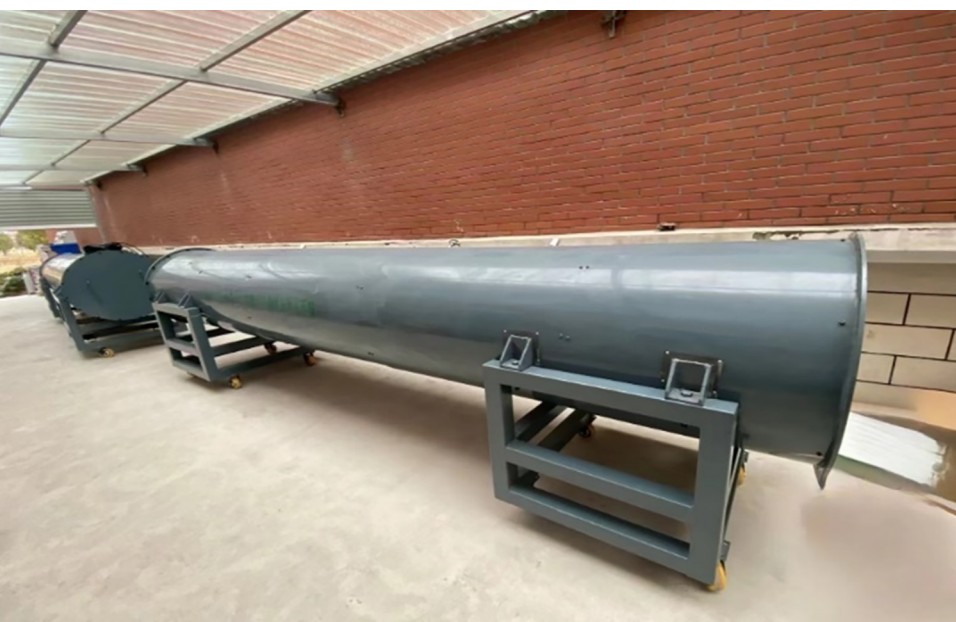

**Fig 2. Simulation platform for porous media in goaf.**

experiment, an initial calibration of the spectrum analyzer was conducted using a standard signal source, ensuring its precise capture and analysis of signals across different frequencies. The sensitivity of the receiver was measured using a known signal source, ensuring that the signals received during the experiment accurately reflect the characteristics of electromagnetic wave propagation. The background noise level of the experimental platform was measured under

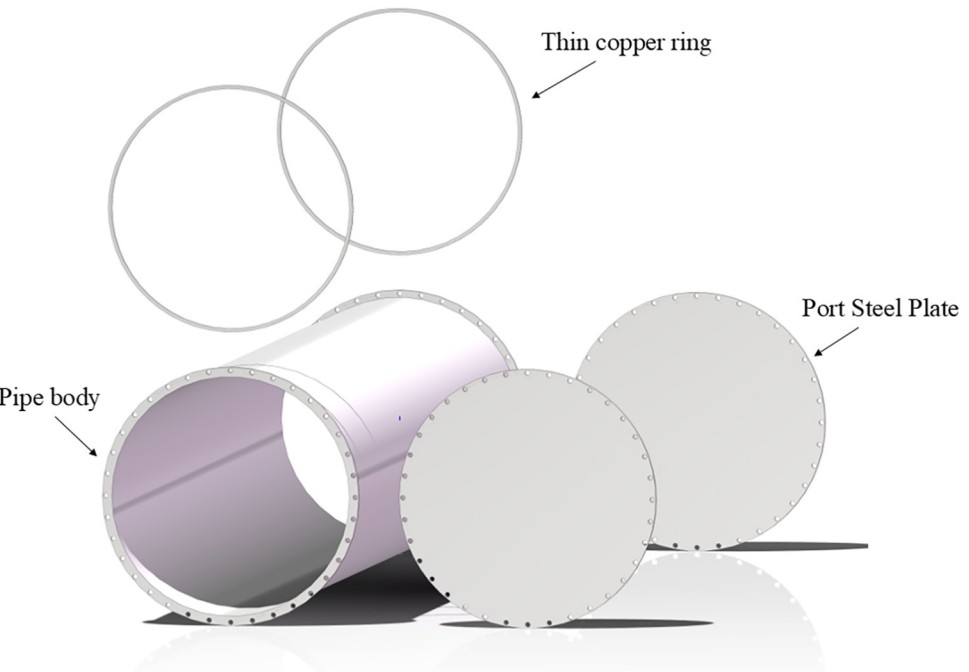

**Fig 3. Interior diagram of goaf simulation platform.**

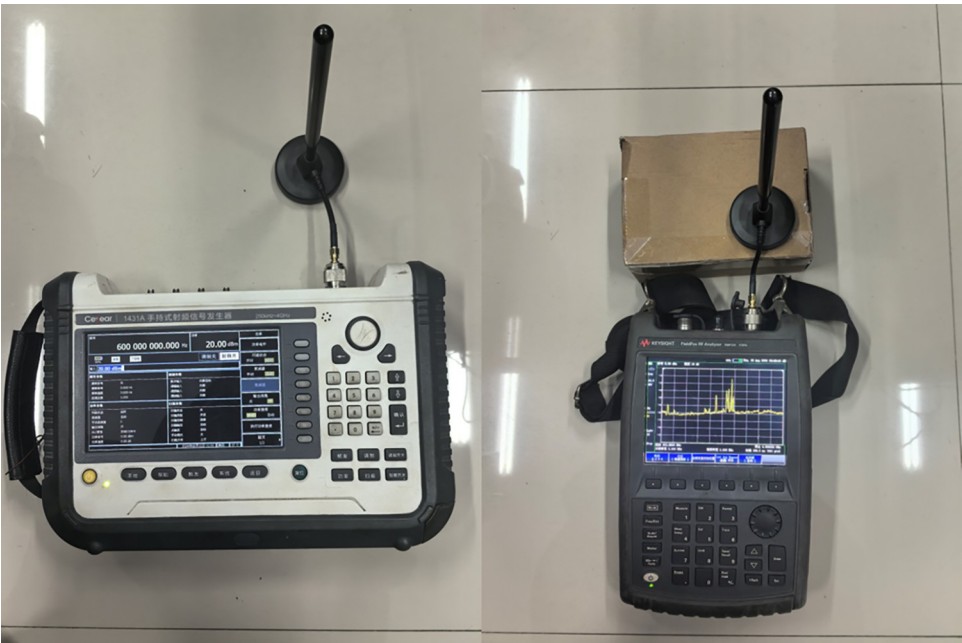

**Fig 4.** RF signal generator (left) RF analyzer (right).

conditions with no signal transmission to eliminate the impact of environmental noise on signal reception.

During this experiment, we employed a high-precision spectrum analyzer to continuously monitor the 600–900 MHz frequency band in real time. The spectrum analyzer was capable of detecting signal strength and identifying interference sources at various frequencies. It also determined whether any signals exceeded the expected frequencies, thereby ensuring the absence of signal mixing in other frequency bands during the experiment. The RF signal generator and RF analyzer are shown in Fig 4.

## 2.3 Experimental error analysis

Variations in the external environment can significantly impact the propagation characteristics of electromagnetic waves. Signal attenuation and refractive index may be altered by fluctuations in temperature and humidity, leading to inconsistent results. These environmental factors alter the dielectric constant of the medium, thereby affecting the attenuation rate and propagation stability of the electromagnetic wave signals.

The spectrum analyzer and signal transmitter are crucial measurement instruments in the experiment. If the measurement accuracy of the instruments is not high, it could result in inaccurate signal strength and frequency data. Common precision errors in equipment include the instability of the signal generator's transmitting power. Fluctuations or errors in the transmitter's power output can cause the actual transmitted signal strength to deviate from the set value, resulting in inaccurate signal strength measurements at the receiving end, thereby affecting the calculation of attenuation and propagation distance.

All experiments conducted in this study were carried out within a closed-loop goaf simulation platform, which maintained stable temperature and humidity to minimize the impact of environmental variations on the experimental results. The spectrum analyzer utilized in our experiments features high resolution and high sampling rates, enabling precise differentiation

of subtle frequency signal variations in the frequency domain, thereby reducing quantization errors in frequency and amplitude, which ensures the accuracy of measurements and generated signals, enhancing the reliability of the results. During the experimental process, to enhance the reliability and accuracy of the results, we employed a method of repeated trials to obtain an average value, thereby reducing the impact of errors and ensuring that the final average is closer to the true measurement value.

# 3 Experimental investigation of electromagnetic wave propagation characteristics in goaf environment

## 3.1 Platform verification experiment with porosity of 1

To validate the effectiveness of the experimental platform and ensure the accuracy of subsequent data, an electromagnetic wave signal transmission experiment with a porosity of 1 was conducted. Throughout the experiment, both the environment and platform parameters were maintained in their default configurations, and potential interference from unrelated factors was systematically excluded. The experimental results are presented in Fig 5.

As illustrated in **Fig 5**, with constant signal transmission power, the signal strength diminishes as the propagation distance increases, attributed to free space path loss. The attenuation patterns of electromagnetic wave intensity with distance are similar across all frequencies. Initially, signal attenuation follows a logarithmic function, during the later phase, the attenuation exhibited a gradual and smooth decline., consistent with the principles of electromagnetic wave propagation in a vacuum. These observations confirm that the signal propagation characteristics experiment conducted on the simulation platform is both feasible and reliable. The signal strength at 600 MHz decreased from −3.35 dBm at 1 meter to −22.69 dBm at 12 meters. At 700 MHz, the signal strength declined from −5.61 dBm at 1 meter to −24.84 dBm at 12 meters. For 800 MHz, the signal intensity reduced from −7.27 dBm at 1 meter to −26.82 dBm at 12 meters. Similarly, at 900 MHz, the signal strength diminished from −8.45 dBm at 1 meter to −29.68 dBm at 12 meters. The attenuation rates for 600 MHz and 700 MHz signals are comparable, with signal intensity remaining relatively strong at 12 meters. The attenuation rate for the 700 MHz signal is slower than that for the 800 MHz and 900 MHz signals, with the 900 MHz signal exhibiting the greatest attenuation. This indicates that as the signal frequency increases, the signal strength at a distance of 12 meters decreases, with a more pronounced difference in signal strength between frequencies.

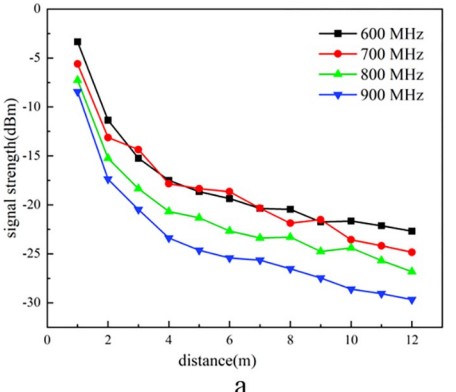 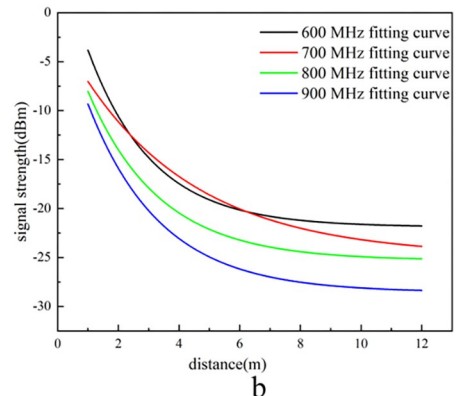

**Fig 5. Signal intensity and fitting curves at porosity 1.**

### 3.2 Platform simulation experiment under porosity of 0.2 and 0.4

The porosity of the goaf typically ranges between 0.1 and 0.4. Due to the limitations of the simulation platform, which does not replicate high ground pressure conditions, achieving a porosity of 0.1 is unfeasible. Additionally, the single-particle size stones used do not adequately represent the variable particle sizes found in a natural goaf. Consequently, this study investigated the propagation characteristics of electromagnetic waves at frequencies when the porosity was set to 0.2 and 0.4. During the experiment, a 1-meter-thick layer of porous medium material was added to the platform, and both ends of the pipe were sealed to ensure system integrity. The signal intensities of 600 MHz, 700 MHz, 800 MHz, and 900 MHz electromagnetic waves, after traversing the medium, were measured. The variation in electromagnetic wave signal intensity with propagation distance at different frequencies is illustrated in Fig 6.

As shown in **Fig 6**, under the porosities of 0.2 and 0.4 on the goaf simulation platform, electromagnetic waves maintain the same transmission power. The received signal strength values for four different frequencies decrease with increasing propagation distance. At equivalent propagation distances, the signal intensity after traversing the porous medium is lower compared to that in an experimental environment with a porosity of 1. This indicates that the signal transmission attenuation is more pronounced when passing through the porous medium. Moreover, in comparison to the control experiment with a porosity of 1, the signal intensity fluctuations through the porous media are more pronounced. This observation suggests that the transmission of electromagnetic waves is significantly impacted by the porous medium environment in the goaf, resulting in pronounced multipath effects and a deterioration in signal quality.

Electromagnetic waves of different frequencies exhibit the same attenuation pattern in environments with porosities of 0.2 and 0.4. For a porosity of 0.2, the signal intensity at 600 MHz attenuates from −6.75 dBm at 1 meter to −52.79 dBm at 12 meters. At a porosity of 0.4, the signal intensity attenuates from −10.47 dBm to −56.09 dBm. At 700 MHz, the signal intensity attenuates from −8.44 dBm to −41.4 dBm when the porosity is 0.2, and from −14.02 dBm to −42.27 dBm when the porosity is 0.4. The attenuation of 600 MHz and 700 MHz signals is lower when traversing the coal-rock mixed porous medium due to their longer wavelengths associated with lower frequencies. As illustrated in **Fig 6**, porosity exerts minimal influence on early attenuation but predominantly impacts the smooth attenuation region in the later phase, with the attenuation rate being directly proportional to the porosity. According to the energy formula for electromagnetic waves [31], higher frequencies correspond to greater initial signal

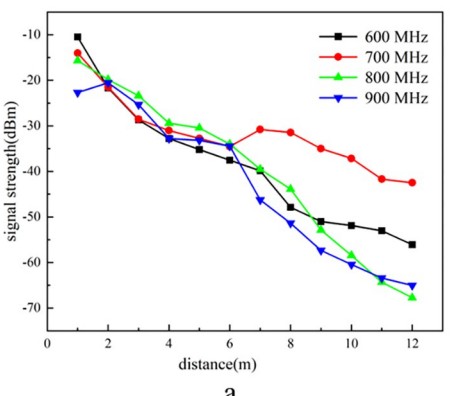
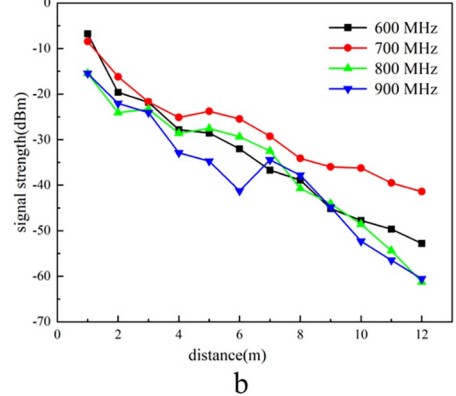

**Fig 6. Signal intensity curves at 0.2 and 0.4 porosity.**

energy. Consequently, the 700 MHz signal demonstrates superior resistance to dielectric loss in the porous media environment. Even after propagating for 12 meters, the signal intensity remains sufficient to meet the transmission requirements for monitoring nodes within the goaf.

At 800 MHz, the signal intensity attenuates from −15.47 dBm to −61.27 dBm when the porosity is 0.2, and from −5.68 dBm to −67.74 dBm when the porosity is 0.4. At 900 MHz, the signal intensity attenuates from −15.45 dBm to −60.56 dBm with a porosity of 0.2, and from −22.68 dBm to −65.01 dBm with a porosity of 0.4. Compared to frequencies of 800 MHz and above, the signal intensities of 600 MHz and 700 MHz are relatively similar at a distance of 12 meters in an environment with a porosity of 1. However, in a porous medium with a porosity of 0.4, the maximum difference in signal intensity can reach up to 23 dBm. These observations indicate that increasing porosity results in greater attenuation amplitude of signal intensity across all frequencies. While porosity does impact the attenuation of low-frequency signals, the effect is relatively minor. As frequency increases, signal attenuation becomes more pronounced, and higher porosity exacerbates this attenuation.

Analysis indicates that higher frequencies correspond to shorter wavelengths, which significantly reduce resistance to dielectric loss and stability. Additionally, increased porosity leads to a more complex pore structure within the porous medium, which enhances the reflection and transmission paths of the electromagnetic wave signal, resulting in increased signal attenuation. According to effective medium theory, the presence of pores alters both the propagation speed and attenuation characteristics of electromagnetic waves within the medium, thereby impacting the stability and variability of the signal.

# 4 Simulation study of electromagnetic wave propagation characteristics in goaf environments

Physical simulation experiments can accurately reflect the inherent properties and internal relationships of systems. However, they involve long experimental cycles and numerous influencing factors. These experiments are constrained to specific conditions and data collection points. Additionally, when multiple experimental variables are involved, the associated costs increase significantly. Numerical simulation technology offers cost-effectiveness and high efficiency, enabling simulation research under a range of complex conditions and comprehensive monitoring of various parameters. Consequently, an electromagnetic wave propagation model for porous media in goaf is developed to investigate the transmission characteristics of electromagnetic waves across different frequencies and porosities.

## 4.1 Coal-rock porous media model construction and parameter setting

By applying suitable boundary conditions, the propagation characteristics of electromagnetic waves in this model are aligned with those in free space. Additionally, a geometric model with an appropriate volume is established to serve as the propagation medium for electromagnetic waves. Following extensive adjustments to the model size, the dimensions of the propagation domain are established as 1.5 m × 1.5 m × 10 m, with a quarter-wavelength dipole antenna positioned at the center of the left side. During the simulation, the antenna is designed to match the emission frequency to ensure consistent emission efficiency. The detailed model is illustrated in Fig 7. The antenna's main body is cylindrical with a characteristic impedance of 50 ohms, and it is excited along the Z-axis with a power of 20 dBm. The antenna surface is defined with an impedance boundary condition, and its electrical properties—such as relative dielectric constant, conductivity, and permeability—are set to those of copper.

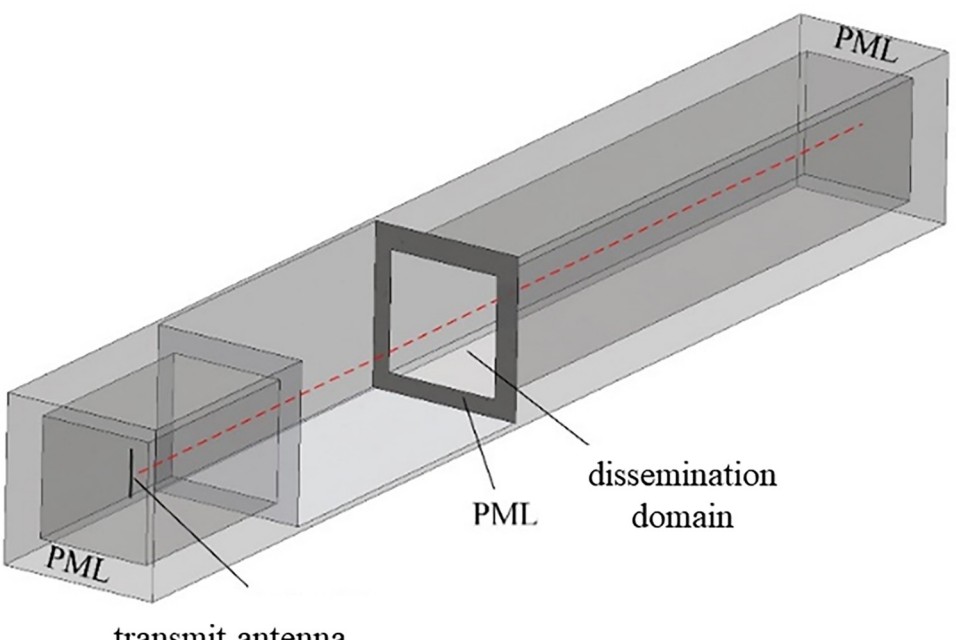

**Fig 7. EM wave propagation model in porous media of goaf.**

When constructing the geometric model of coal-rock mixed media in the coal mine goaf, two essential simplification steps are required: first, simplifying the shape by replacing irregular geometric structures with more regular designs, thereby greatly enhancing computational efficiency. The second step involves simplifying particle sizes by adjusting individual particle parameters and constructing a model with porosity similar to that of the goaf. This approach can notably decrease computational complexity and subsequently alleviate workload. Consequently, four types of regular icosahedrons were designed, each with external sphere diameters of 5 cm, 7 cm, 8 cm, and 9 cm. Porosity was evaluated based on the volume and quantity of these structures, resulting in calculated porosity values of 0.1, 0.2, 0.3, and 0.4 for the respective icosahedrons. The porosity model is illustrated in **Fig 8**.

The properties of the pores in the porous medium are equivalent to those of air. The remaining materials replicate the medium properties of the electromagnetic wave signal transmission experimental platform in the goaf. The dielectric constants and resistivities of various media materials are detailed in Table 1 [32–34].

## 4.2 Numerical simulation of the influence of porosity of porous media on electromagnetic wave propagation law

The porosity of the coal-rock mixed porous media is set to 0.1, 0.2, 0.3, and 0.4, with the resistivity of the coal-rock medium fixed at 6 and the relative dielectric constant set to 4. The study investigates the propagation characteristics of electromagnetic waves at frequencies of 600, 700, 800, and 900 MHz within the porous media environment. The simulation results in the simulation software are shown in Fig 9.

The frequency curves for 800 MHz and 900 MHz exhibit significant fluctuations, making it challenging to intuitively discern the attenuation trend of the signal strength. Consequently, the data were smoothed using the adjacent averaging method due to the large volume of coordinate data and the uneven distribution of the abscissa. A window size of 1000 points was applied. The processed results are presented in Fig 10.

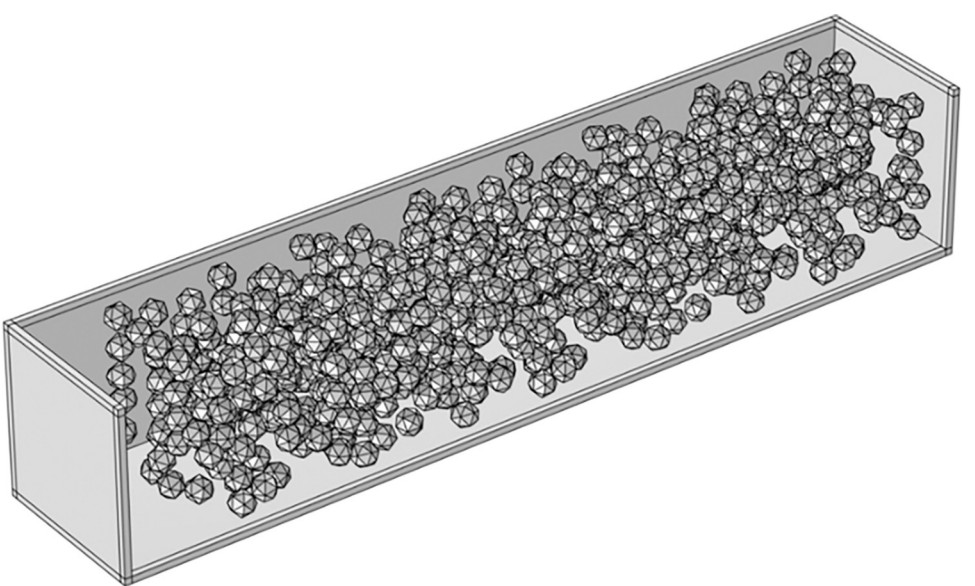

**Fig 8. Porosity model.**

As illustrated in Figs 9 and 10, signal intensity varies markedly with frequency and porosity, with distinct attenuation patterns observed across different porosities. To mitigate potential errors associated with signal data at the extreme frequencies, the intermediate frequency of 700 MHz is selected for analysis. This choice enhances the reliability of the results and ensures their broader applicability. At a porosity of 0.2, the signal intensity of the 700 MHz signal is attenuated to -41 dBm at 10 meters. In comparison, at a porosity of 0.4, the signal intensity is attenuated to -55 dBm at the same distance. The simulation results were compared with the experimental results from the platform. For a porosity of 0.2, the signal intensities were in close agreement. Although some discrepancies were observed at a porosity of 0.4, the differences fall within the anticipated error range. The experimental results obtained from the simulation platform are susceptible to environmental and experimental conditions, leading to greater attenuation compared to the simulation data at the same frequency. However, at porosities of 0.2 and 0.4, the experimental results align with the overall attenuation trends observed in the simulation data. This consistency demonstrates that the electromagnetic wave propagation model for porous media in the goaf is accurate and effectively applicable across the tested propagation ranges.

At identical transmission power levels, high-frequency electromagnetic waves inherently contain more energy, resulting in higher signal strength. However, when propagating to a distance of 1 meter at the same porosity, the signal intensity of higher-frequency electromagnetic waves is lower compared to that of lower-frequency waves. This observation indicates that the attenuation rate in the initial logarithmic attenuation phase is positively correlated with frequency, meaning that higher-frequency electromagnetic waves experience greater attenuation

**Table 1. Dielectric constant and conductivity of different coals.**

| Material | Dielectric constant | Resistivity ($k\Omega{\cdot}m$) |
| --- | --- | --- |
| Quartzite | 6.5 | 0.9 |
| Sandstone | 4.6 | 5.4 |
| Granite | 2.5 | 6.5 |

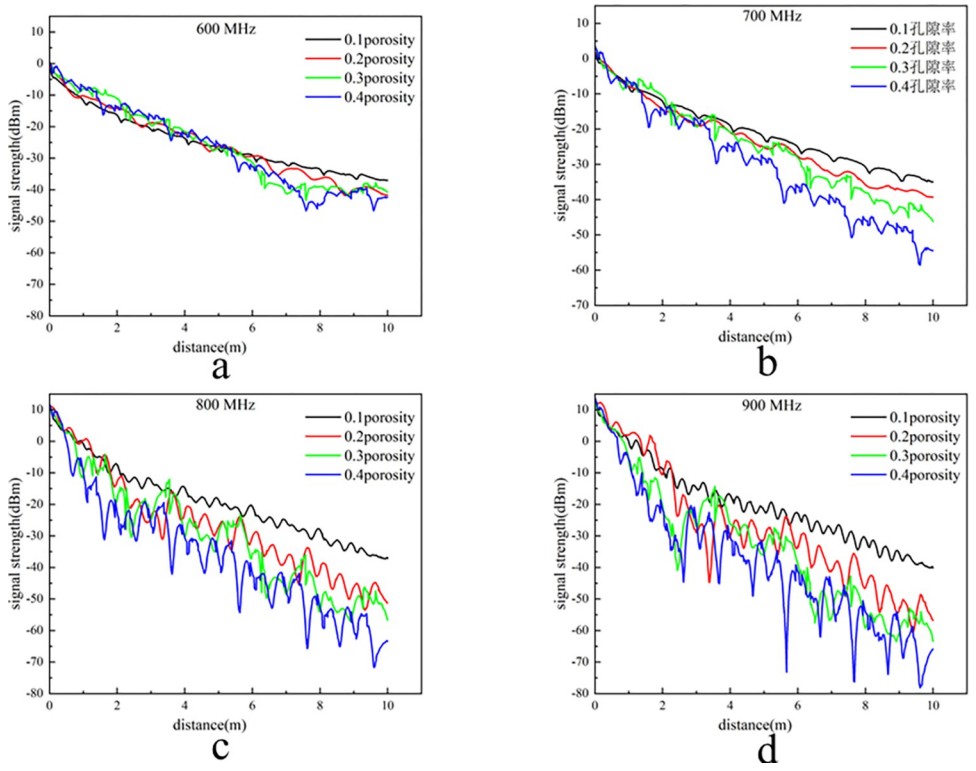

**Fig 9. Signal intensity curve at different porosities.**

in the early propagation stage. As illustrated in Figs 9 and 10, the signal intensity at 600 MHz remains largely consistent across different porosities within the first 5 meters. At 700 MHz, the signal intensity is consistent for the first 3 meters, while at 800 MHz and 900 MHz, the intensity only remains consistent within the first meter. As frequency increases, the wavelength of the signal shortens, making it more susceptible to absorption by the medium in a porous environment. This absorption converts electromagnetic energy into heat or other forms of energy, leading to a higher rate of signal attenuation with increasing frequency. Additionally, higher frequency electromagnetic waves experience more rapid phase changes per unit distance, causing more pronounced fluctuations in signal intensity during propagation.

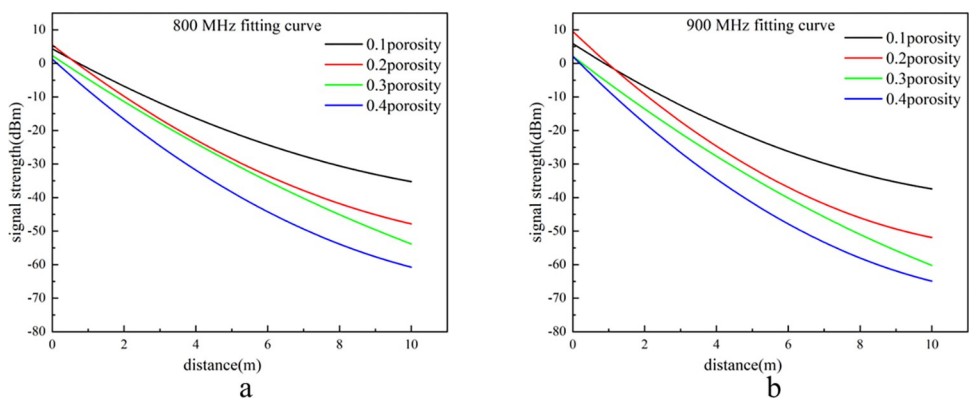

**Fig 10. Processed intensity curve of 800 MHz and 900 MHz.**

As illustrated in Fig 9(A), for a frequency of 600 MHz: at a porosity of 0.1, the signal strength decreases rapidly in the initial stage and then stabilizes as the propagation distance increases. As the porosity increases to 0.2, 0.3, and 0.4, the rate of signal attenuation and volatility both increase progressively, although the signal strength still stabilizes with increasing propagation distance. By comparing the signal attenuation values for frequencies of 600 MHz, 700 MHz, 800 MHz, and 900 MHz as shown in Figs 9 and 10, it is evident that the impact of porosity on electromagnetic wave attenuation is frequency-dependent. Specifically, higher frequencies exhibit a greater sensitivity to porosity, resulting in increased attenuation. At a porosity of 0.1, the signal attenuation values for frequencies of 600 MHz, 700 MHz, 800 MHz, and 900 MHz are 35 dBm, 40 dBm, 46 dBm, and 53 dBm, respectively. At a porosity of 0.2, the corresponding attenuation values are 41 dBm, 42 dBm, 61 dBm, and 68 dBm, respectively. As the porosity increases to 0.3 and 0.4, signal attenuation at the same frequency becomes progressively more pronounced. Analysis reveals that higher porosity results in greater irregularity in the size, shape, and distribution of internal pores, leading to increased scattering and diffraction of electromagnetic waves. Consequently, electromagnetic waves experience more significant phase changes and fluctuations during propagation, which enhances signal attenuation and distortion, thereby increasing both the rate of attenuation and the variability of the signal. Based on the preceding analysis, it is evident that, to optimize communication distance and signal quality within the goaf, utilizing the minimum number of sensor nodes while ensuring maximum coverage is crucial. The selection of a transmission frequency of 700 MHz is thus recommended, as it effectively meets the communication requirements of the goaf environment.

## 4.3 Influence of pore structure characteristics of porous media on electromagnetic wave propagation and analysis of result applicability

The pore structure characteristics of the goaf, including pore size distribution and connectivity, play a critical role in determining the electromagnetic wave propagation properties within this region. The goaf represents a complex network of cavities and fractures formed as a result of post-mining activities. The pore structure within this region is highly intricate, resulting from the deformation and destruction of the surrounding rock layers. The size, connectivity, and spatial distribution of the pore structure significantly influence electromagnetic wave behaviors, including reflection, attenuation, scattering, and multipath effects [35,36].

Micro-pores, ranging from microns to millimeters, are predominantly composed of fine cracks between rock particles or debris generated during mining activities. The size of these pores typically remains smaller than the wavelength of electromagnetic waves, particularly at lower frequencies. These micro-pores primarily influence high-frequency electromagnetic waves (e.g., those in the GHz range), leading to significant signal scattering and rapid attenuation. Medium-sized pores (ranging from millimeters to centimeters) are formed through processes such as rock fracturing and localized collapse, and are widely distributed throughout the region. Medium pores can induce signal reflection or partial penetration, which in turn leads to multipath effects and enhances the complexity of signal propagation pathways. Large voids, typically meters in scale or larger, represent the core structural elements of the goaf, formed as a result of extensive rock collapse or excavation. These large voids exert a substantial influence on the propagation of low-frequency electromagnetic waves (e.g., at or below 300 MHz), facilitating long-distance signal propagation while simultaneously increasing the likelihood of multipath effects.

Pore connectivity refers to the extent of interconnection between the various pores within the goaf, and the intricate network structure formed as a result. Pore connectivity is influenced

by factors such as the mining process, rock fracturing, and collapse, all of which directly impact the propagation paths of electromagnetic waves within the goaf. When the pores within the goaf are well interconnected, they form a highly complex three-dimensional network structure. These interconnected pores can offer multiple propagation pathways for electromagnetic waves, leading to more pronounced multipath effects. In such cases, the electromagnetic waves propagate along multiple distinct paths, with signals potentially reaching the receiver through different routes, leading to either signal interference or enhancement. If pore connectivity is poor, this implies that propagation paths between pores are limited or obstructed. Under such conditions, electromagnetic waves may be confined to specific propagation paths, making them susceptible to rapid signal attenuation or blockage.

The experimental results presented are derived from studies conducted under specific porosity conditions, and these results may have limitations in certain specialized coal mining environments, where geological conditions are among the primary factors influencing electromagnetic wave propagation. There are significant differences in geological characteristics, such as depth and rock type, among different coal mines. These differences directly impact the attenuation and reflection characteristics of electromagnetic wave propagation.

Deep coal mines typically exhibit higher rock pressure and density, coupled with relatively lower porosity, resulting in greater resistance to electromagnetic wave propagation and faster signal attenuation. In contrast, shallow coal mines generally have higher porosity, leading to more pronounced reflection and multipath effects during electromagnetic wave propagation. Therefore, electromagnetic wave propagation models and optimization strategies for deep and shallow coal mines must be adjusted accordingly, and the applicability of current research findings in these distinct environments requires further validation.

The type of rock present in the goaf of coal mines significantly influences the characteristics of electromagnetic wave propagation. Ore bodies containing high concentrations of metallic minerals may lead to greater electromagnetic wave losses during propagation, whereas sedimentary rocks with complex pore structures are more likely to cause increased reflection and scattering. These differences in rock properties suggest that current propagation models may not be universally applicable across all mining environments. In future research, we will further validate the adaptability of the results through field tests and additional numerical simulations to enhance their broader applicability.

## 5. Conclusions

To accurately capture the electromagnetic wave propagation characteristics in porous media within the goaf, a dedicated simulation experiment platform was developed, and an electromagnetic wave propagation model for porous media in the goaf was established. Experiments were conducted to investigate electromagnetic wave propagation at various frequencies within this goaf environment. The findings are summarized as follows:

(1) The experimental results obtained from the platform align closely with the signal attenuation trends observed in the simulation results. For propagation distances less than 4 meters, both the simulation platform and software show that the signal intensity curves for different frequencies converge at porosities of 0.2 and 0.4.

(2) The attenuation rate of electromagnetic waves in the initial logarithmic attenuation phase exhibits a positive correlation with frequency, increasing frequency enhances the multipath effect on the pore structure, leading to a greater attenuation amplitude of high-frequency electromagnetic waves during the initial phase. Porosity primarily influences the smooth attenuation zone in the later stages, with the rate of signal intensity attenuation being directly proportional to the porosity.

(3) As the frequency of electromagnetic waves rises, the initial energy of the waves correspondingly increases. Nevertheless, at elevated frequencies, the attenuation of signal strength becomes significantly more pronounced, with the magnitude of attenuation exceeding the rate of increase in initial energy. This phenomenon is further amplified with increasing porosity.

(4) Under various experimental and simulation conditions, electromagnetic waves at a frequency of 700 MHz exhibit a relatively low attenuation coefficient and demonstrate considerable stability and reliability. Electromagnetic waves within this frequency band effectively mitigate signal attenuation over long distances, making this frequency range an optimal choice for enhancing the performance of wireless sensor networks.

## Supporting information

**S1 Dataset. Numerical simulation data.**
(XLSX)

**S2 Dataset. Simulation platform experimental data.**
(XLSX)

## Acknowledgments

Thanks to the suggestions made by teacher Qingsong Zhang and Hui Zhuo during the research and the revision and editing of the article when it is completed.

## Author Contributions

**Data curation:** Qingsong Zhang, Hui Zhuo.

**Formal analysis:** Weikang Zhao, Hui Zhuo.

**Funding acquisition:** Long Lin.

**Investigation:** Zuojian Li.

**Methodology:** Qingsong Zhang.

**Project administration:** Hui Zhuo, Long Lin.

**Software:** Weikang Zhao, Long Lin.

**Supervision:** Qingsong Zhang, Weikang Zhao, Hui Zhuo, Zuojian Li.

**Validation:** Hui Zhuo.

**Writing – original draft:** Qingsong Zhang, Weikang Zhao.

**Writing – review & editing:** Hui Zhuo.

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
