## [Decision Letter · Decision Letter 0]

17 Oct 2024

PONE-D-24-39974Investigation of Electromagnetic Wave Propagation Characteristics Across Various Frequencies in Porous Media of GoafPLOS ONE

Dear Dr. zhuo,

Thank you for submitting your manuscript to PLOS ONE. After careful consideration, we feel that it has merit but does not fully meet PLOS ONE’s publication criteria as it currently stands. Therefore, we invite you to submit a revised version of the manuscript that addresses the points raised during the review process. Please carefully address the reviewers' questions, especially those of reviewer #2, who requested more substantial changes.

We look forward to receiving your revised manuscript.

Kind regards,

Marko Čanađija

Academic Editor

PLOS ONE

Journal Requirements:

2. Thank you for stating the following financial disclosure: “National Natural Science Foundation of China (52204192), Anhui Provincial Key Research and Development Project (2022m07020006)”.

3. We note that your Data Availability Statement is currently as follows: “All relevant data are within the manuscript and in Supporting Information files.”

Please confirm at this time whether or not your submission contains all raw data required to replicate the results of your study. Authors must share the “minimal data set” for their submission. PLOS defines the minimal data set to consist of the data required to replicate all study findings reported in the article, as well as related metadata and methods (https://journals.plos.org/plosone/s/data-availability#loc-minimal-data-set-definition). For example, authors should submit the following data: - The values behind the means, standard deviations and other measures reported; - The values used to build graphs; - The points extracted from images for analysis. Authors do not need to submit their entire data set if only a portion of the data was used in the reported study. If your submission does not contain these data, please either upload them as Supporting Information files or deposit them to a stable, public repository and provide us with the relevant URLs, DOIs, or accession numbers. For a list of recommended repositories, please see https://journals.plos.org/plosone/s/recommended-repositories. If there are ethical or legal restrictions on sharing a de-identified data set, please explain them in detail (e.g., data contain potentially sensitive information, data are owned by a third-party organization, etc.) and who has imposed them (e.g., an ethics committee). Please also provide contact information for a data access committee, ethics committee, or other institutional body to which data requests may be sent. If data are owned by a third party, please indicate how others may request data access.

5. Please ensure that you refer to Figure 1 in your text as, if accepted, production will need this reference to link the reader to the figure.

7. Please remove your figures from within your manuscript file, leaving only the individual TIFF/EPS image files, uploaded separately. These will be automatically included in the reviewers’ PDF**.**

Reviewers' comments:

Reviewer's Responses to Questions

**Comments to the Author**

1. Is the manuscript technically sound, and do the data support the conclusions?

Reviewer #1: Yes

Reviewer #2: Yes

2. Has the statistical analysis been performed appropriately and rigorously? 

Reviewer #1: Yes

Reviewer #2: Yes

3. Have the authors made all data underlying the findings in their manuscript fully available?

Reviewer #1: No

Reviewer #2: Yes

4. Is the manuscript presented in an intelligible fashion and written in standard English?

Reviewer #1: Yes

Reviewer #2: Yes

5. Review Comments to the Author

Reviewer #1: This article reveals the characteristics of EM signal attenuation at different frequencies in their lab. I think the conclusions of this article might provide a reference for field work。 But I think there are some part of manuscript should be reconcideration:

1.Line 39, I think it’s ‘goaf’, not ‘goafs’;

2. Line 64, I think the authors should explain why the propagation distance is critical.

3. Line 72, I don’t think it can provide a theoretical foundation, because the equations never seen in this paper.

4 Line 107 How to make sure the frequency range in 600~900Mhz without 1G or 2G Hz?

5. Section 1.2 I advise to add an internal schematic figure to the platform.

6. Line 533, please revise the figure name form. I am confused about the figure(b), which component to fit?

Reviewer #2: The author has conducted a study that establishes an experimental platform for the propagation of electromagnetic wave signals in goaf areas and develops a propagation model for electromagnetic waves in porous media within the areas. Through experimental and numerical simulation methods, the study investigates the propagation characteristics of electromagnetic waves at different frequencies in the porous medium environment of goaf. The paper is well-structured with a solid logical flow. The images and tables are clear, reflecting the extensive work done by the author. However, some issues need to be addressed before acceptance:

1. It is recommended that the author expand the literature review, particularly regarding recent advancements in the study of electromagnetic wave propagation characteristics in coal mine goaf.

2. The author should provide more detailed information on the experimental design and methodologies, including the construction of the experimental platform, the calibration process of measurement devices, and the parameter settings used during the experiments.

3. While the article mentions the porosity of the porous medium, it lacks a detailed description of other pore structure characteristics, such as pore size distribution and pore connectivity. The impact of the characteristics on electromagnetic wave propagation warrants further discussion.

4. The data analysis section lacks a discussion on the sources of error. The author should clarify potential sources of error in the experiments and analyze how the errors may influence the results.

5. The applicability of the results presented in the article to different coal mine environments needs further exploration.

6. PLOS authors have the option to publish the peer review history of their article (what does this mean?). If published, this will include your full peer review and any attached files.

Reviewer #1: No

Reviewer #2: No

---

## [Author Response · Author response to Decision Letter 0]

2 Nov 2024

Dear Editor:

Thank you for your letter and for the reviewers’ comments concerning our manuscript entitled “Investigation of Electromagnetic Wave Propagation Characteristics Across Various Frequencies in Porous Media of Goaf”.( Manuscript Number: PONE-D-24-39974) Those comments are all valuable and very helpful for revising and improving our paper, as well as the important guiding significance to our researches. We have studied comments carefully and have made correction which we hope meet with approval. The main corrections in the paper and the responds to the reviewer’s comments are as follows.

Best regards

Dr. Hui Zhuo on the behalf of the authors

Responds to the Editor comments:

1. We have revised the manuscript using the provided template in accordance with the formatting requirements of PLOS ONE.

2. We have amended the funding statement in the manuscript to clarify the involvement of the funders in the research design and analysis data stages.

3. All relevant data have been uploaded as supporting information files at the end of the Word document. If readers require access to these data, they can be provided upon request.

4. We have created and verified the corresponding author's ORCID iD and updated it in the system.

5. We have ensured that Figure 1 is correctly cited in the text.

6. We have added descriptions to the supplementary information files and updated the references in the manuscript.

7. We have removed the figures from the manuscript and uploaded the TIFF format files separately.

Responds to the reviewer’s comments:

Reviewer #1:

Issue

1.Line 39, I think it’s ‘goaf’, not ‘goafs’;

Responds：Thanks for the reviewer 's opinion. We thank the reviewer for their valuable feedback. We concur with your assessment and have made the necessary revisions in the updated draft. We appreciate your guidance once again. Line 31、Line 37、Line 78、Line 97、Line 100

2. Line 64, I think the authors should explain why the propagation distance is critical.

Responds：We sincerely appreciate the reviewer’s insightful suggestions. In response to Line 64, we have incorporated a detailed explanation regarding the significance of propagation distance in the revised manuscript.

The goaf is characterized by complex terrain, numerous obstacles, and significant spatial limitations. Within this context, electromagnetic wave propagation experiences rapid attenuation due to environmental interference. Consequently, the effective transmission distance of information via wireless communication is a critical determinant of the system's practicality. Should the spacing of wireless communication be inadequate, additional monitoring equipment will be required to comprehensively cover the goaf, thereby escalating the costs associated with the monitoring system layout and complicating system maintenance and management. The long-distance propagation of wireless signals within the goaf not only ensures extensive communication coverage but also significantly enhances the system's overall efficiency. We sincerely appreciate. Page 4- Page 5

3. Line 72, I don’t think it can provide a theoretical foundation, because the equations never seen in this paper.

Responds：We sincerely appreciate the reviewer’s insightful suggestions. We have enhanced the discussion in this section by incorporating additional background information on relevant theories. Furthermore, we have explicitly delineated the applications and sources of the equations referenced in this paper. We have supplemented the relevant references to facilitate readers’ comprehension of the theoretical foundations of these equations.

Based on the selected frequency range of 600-900 MHz, and applying wave theory and transmission line theory, the cutoff frequency of the pipeline can be calculated using the formula:

█(λ=C/f_c )(1)

For a circular waveguide, f_c=600 MHz,C≈3×10^8,the relationship between the cutoff wavelength and the diameter is as follows：

█(λ=(2.405×D)/1)(2)

Calculated according to the equation：

D≈0.208m≈20.8cm

According to calculations, the diameter of a circular pipeline with a cutoff frequency below 600 MHz is approximately 0.208 meters. To facilitate the experiment, the pipeline diameter was increased to 0.75 meters. This adjustment not only provides additional tolerance but also helps avoid potential issues arising from non-compliance with design standards. Therefore, the main structure of the platform was designed as a metal steel pipe with an inner diameter of 0.75 m and a length of 5 m. Page 7

4 Line 107 How to make sure the frequency range in 600~900Mhz without 1G or 2G Hz?

Responds：We appreciate the reviewer's suggestion and concur with their assessment. During this experiment, we employed a high-precision spectrum analyzer to continuously monitor the 600-900 MHz frequency band in real time. The spectrum analyzer was capable of detecting signal strength and identifying interference sources at various frequencies. It also determined whether any signals exceeded the expected frequencies, thereby ensuring the absence of signal mixing in other frequency bands during the experiment. Page 9

5. Section 1.2 I advise to add an internal schematic figure to the platform.

Responds：We appreciate the reviewer's suggestion. In the revised draft, we have included the relevant internal structure diagram to enable readers to better understand the design and working principle of the platform. The newly added schematic is shown below.

Fig. 3 Schematic diagram of the interior of the goaf simulation platform

Line 177

6. Line 533, please revise the figure name form. I am confused about the figure(b), which component to fit?

Responds：We appreciate the reviewer's comments. Regarding Line 533, Figure (a) illustrates the signal intensity attenuation curves at different frequencies with a porosity of 0.2, while Figure (b) depicts the corresponding curves at a porosity of 0.4. We have revised the figure titles according to your suggestions to ensure greater clarity. Thank you for your feedback; we will strive to make the illustrations clearer and more comprehensible. Line 252

Reviewer #2:

Issue

1. It is recommended that the author expand the literature review, particularly regarding recent advancements in the study of electromagnetic wave propagation characteristics in coal mine goaf.

Thank you very much for your invaluable input on our manuscript. Based on your suggestions, we have expanded the literature review section, with particular emphasis on recent advancements in the study of electromagnetic wave propagation characteristics in goaf areas. The revised review not only covers the major research achievements in this field over recent years but also delves into the propagation characteristics of electromagnetic waves across different frequency bands in complex geological conditions within goaf areas, the application of numerical simulation methods (such as ray tracing and finite-difference time-domain methods), and the progress of related experimental studies.. We sincerely appreciate! Page 3- Page 4

2. The author should provide more detailed information on the experimental design and methodologies, including the construction of the experimental platform, the calibration process of measurement devices, and the parameter settings used during the experiments.

Responds：We sincerely thank the reviewers for their insightful and valuable suggestions. We fully recognize the importance of providing a more comprehensive account of the experimental design and methodological details. In the revised manuscript, we have included a thorough description of the experimental platform construction, the calibration process of the measurement equipment, and the experimental parameter settings, to ensure that readers gain a more complete understanding of the experimental process. We will strive to provide more detailed information in these areas, and we are grateful for your suggestions for improvement. Based on your comments, the modifications we have made are as follows:

The experimental platform primarily consists of a metal pipe structure designed to simulate the electromagnetic wave propagation environment within a goaf. The specific construction process involves: Based on theoretical calculations, the core structure of the experimental platform consists of three steel pipes, each 5 meters in length with an inner diameter of 0.75 meters. The pipeline is filled with a mixed medium of granite and quartzite to simulate the goaf environment., with measurements recorded at 1-meter intervals. The porosity is controlled for each meter along the length of the simulation platform, thereby simulating the effects of varying pore sizes and rock types in a real goaf on electromagnetic wave propagation. Electromagnetic signal transmitters and spectrum analyzers are positioned at both ends of the pipeline. The transmitting antenna is connected to a signal generator to emit electromagnetic wave signals across various frequency bands. The receiving antenna captures the transmitted signal and measures its attenuation and scattering characteristics.

Ensuring the accuracy of the measured data during the experiment is critically dependent on the calibration of the spectrum analyzer and signal generator. Prior to the formal experiment, an initial calibration of the spectrum analyzer was conducted using a standard signal source, ensuring its precise capture and analysis of signals across different frequencies. The sensitivity of the receiver was measured using a known signal source, ensuring that the signals received during the experiment accurately reflect the characteristics of electromagnetic wave propagation. The background noise level of the experimental platform was measured under conditions with no signal transmission to eliminate the impact of environmental noise on signal reception. We sincerely appreciate your meticulous feedback and guidance! Page 7- Page 9

3. While the article mentions the porosity of the porous medium, it lacks a detailed description of other pore structure characteristics, such as pore size distribution and pore connectivity. The impact of the characteristics on electromagnetic wave propagation warrants further discussion.

Responds：We sincerely thank the reviewers for their valuable comments. In addition to porosity, other pore structure characteristics, such as pore size distribution and pore connectivity, also significantly impact electromagnetic wave propagation. We have included additional discussion and analysis on this aspect in the revised manuscript. Based on your suggestions, we have made the following modifications:

The pore structure characteristics of the goaf, including pore size distribution and connectivity, play a critical role in determining the electromagnetic wave propagation properties within this region. The goaf represents a complex network of cavities and fractures formed as a result of post-mining activities. The pore structure within this region is highly intricate, resulting from the deformation and destruction of the surrounding rock layers. The size, connectivity, and spatial distribution of the pore structure significantly influence electromagnetic wave behaviors, including reflection, attenuation, scattering, and multipath effects.

Micro-pores, ranging from microns to millimeters, are predominantly composed of fine cracks between rock particles or debris generated during mining activities. The size of these pores typically remains smaller than the wavelength of electromagnetic waves, particularly at lower frequencies. These micro-pores primarily influence high-frequency electromagnetic waves (e.g., those in the GHz range), leading to significant signal scattering and rapid attenuation. Medium-sized pores (ranging from millimeters to centimeters) are formed through processes such as rock fracturing and localized collapse, and are widely distributed throughout the region. Medium pores can induce signal reflection or partial penetration, which in turn leads to multipath effects and enhances the complexity of signal propagation pathways. Large voids, typically meters in scale or larger, represent the core structural elements of the goaf, formed as a result of extensive rock collapse or excavation. These large voids exert a substantial influence on the propagation of low-frequency electromagnetic waves (e.g., at or below 300 MHz), facilitating long-distance signal propagation while simultaneously increasing the likelihood of multipath effects.

Pore connectivity refers to the extent of interconnection between the various pores within the goaf, and the intricate network structure formed as a result. Pore connectivity is influenced by factors such as the mining process, rock fracturing, and collapse, all of which directly impact the propagation paths of electromagnetic waves within the goaf. When the pores within the goaf are well interconnected, they form a highly complex three-dimensional network structure. These interconnected pores can offer multiple propagation pathways for electromagnetic waves, leading to more pronounced multipath effects. In such cases, the electromagnetic waves propagate along multiple distinct paths, with signals potentially reaching the receiver through different routes, leading to either signal interference or enhancement. If pore connectivity is poor, this implies that propagation paths between pores are limited or obstructed. Under such conditions, electromagnetic waves may be confined to specific propagation paths, making them susceptible to rapid signal attenuation or blockage. This paper primarily investigates the impact of porosity on electromagnetic wave propagation. In our future research, we will explore the effects of pore size distribution and pore connectivity on electromagnetic wave behavior. Thank you once again. Page 22- Page 24

4. The data analysis section lacks a discussion on the sources of error. The author should clarify potential sources of error in the experiments and analyze how the errors may influence the results.

Responds：We appreciate the reviewer's insightful suggestions. In the revised manuscript, we have incorporated a detailed analysis of potential error sources in the experiment, including variations in external environmental conditions and the precision of experimental apparatus. Furthermore, we have examined the potential effects of these errors on experimental outcomes and assessed their implications for the reliability and accuracy of our overall conclusions. Based on your comments, the modifications we have made are as follows:

Variations in the external environment can significantly impact the propagation characteristics of electromagnetic waves. Signal attenuation and refractive index may be altered by fluctuations in temperature and humidity, leading to inconsistent results. These environmental factors alter the dielectric constant of the medium, thereby affecting the attenuation rate and propagation stability of the electromagnetic wave signals.

The spectrum analyzer and signal transmitter are crucial measurement instruments in the experiment. If the measurement accuracy of the instruments is not high, it could result in inaccurate signal strength and frequency data. Common precision errors in equipment include the instability of the signal generator's transmitting power. Fluctuations or errors in the transmitter's power output can cause the actual transmitted signal strength to deviate from the set value, resulting in inaccurate signal strength measurements at the receiving end, thereby affecting the calculation of attenuation and propagation distance.

All experiments conducted in this study were carried out within a closed-loop goaf simulation platform, which maintained stable temperature and humidity to minimize the impact of environmental variations on the experimental results. The spectrum analyzer utilized in our experiments features high resolution and high sampling rates, enabling precise differentiation of subtle frequency signal variations in the frequency domain, thereby reducing quantization errors in frequency and amplitude, which ensures the accuracy of measurements and generated signals, enhancing the reliability of the results. During the experimental process, to enhance the re

---

## [Decision Letter · Decision Letter 1]

26 Nov 2024

PONE-D-24-39974R1Investigation of Electromagnetic Wave Propagation Characteristics Across Various Frequencies in Porous Media of GoafPLOS ONE

Dear Dr. zhuo,

Thank you for submitting your manuscript to PLOS ONE. After careful consideration, we feel that it has merit but does not fully meet PLOS ONE’s publication criteria as it currently stands. Therefore, we invite you to submit a revised version of the manuscript that addresses the points raised during the review process.

We look forward to receiving your revised manuscript.

Kind regards,

Marko Čanađija

Academic Editor

PLOS ONE

Journal Requirements:

**Additional Editor Comments:**

Dear Prof. Zhuo,

almost all issues have been solved, but the reviewers still asks:

"Many format errors, eg. references, format, et al.. Pls correct them. the other issues have been corrected."

Thus, please correct these smaller issues before the manuscript is accepted for publication.

Marko Čanađija

Reviewers' comments:

Reviewer's Responses to Questions

**Comments to the Author**

1. If the authors have adequately addressed your comments raised in a previous round of review and you feel that this manuscript is now acceptable for publication, you may indicate that here to bypass the “Comments to the Author” section, enter your conflict of interest statement in the “Confidential to Editor” section, and submit your "Accept" recommendation.

Reviewer #2: All comments have been addressed

2. Is the manuscript technically sound, and do the data support the conclusions?

Reviewer #2: Partly

3. Has the statistical analysis been performed appropriately and rigorously? 

Reviewer #2: No

4. Have the authors made all data underlying the findings in their manuscript fully available?

Reviewer #2: Yes

5. Is the manuscript presented in an intelligible fashion and written in standard English?

Reviewer #2: Yes

6. Review Comments to the Author

Reviewer #2: Many format errors, eg. references, format, et al.. Pls correct them. the other issues have been corrected.

7. PLOS authors have the option to publish the peer review history of their article (what does this mean?). If published, this will include your full peer review and any attached files.

Reviewer #2: No

---

## [Author Response · Author response to Decision Letter 1]

30 Nov 2024

Thank you for reviewing our manuscript and providing us with valuable feedback. We have carefully examined and revised all relevant sections of the manuscript, addressing each of the comments raised by you and the reviewers. Our specific responses are as follows:

1.Completeness and Accuracy of References

 We have thoroughly reviewed the reference list to ensure its completeness and accuracy.

2.Formatting of References

We have revised the references to comply with the journal’s formatting guidelines. This includes correcting the details of author names, publication years, titles, journal names, volume numbers, and page ranges. All references have been double-checked to ensure there are no omissions or errors.

3.Other Formatting Issues

In accordance with the reviewers’ suggestions, we have further adjusted the overall formatting of the manuscript. This includes paragraph indentation, heading styles, font consistency, and line spacing to fully meet the journal’s requirements.

4.Additional Notes

All modifications made during this revision process have been highlighted in the manuscript and detailed in the response letter. We hope these revisions address your concerns. Should there be any further feedback or additional requirements, we would be more than happy to make the necessary changes.

Once again, we sincerely thank you and the reviewers for your diligent efforts and constructive feedback on our manuscript. Please feel free to contact us if there are any further questions or suggestions.

---

## [Editor Report · Decision Letter 2]

3 Dec 2024

Investigation of Electromagnetic Wave Propagation Characteristics Across Various Frequencies in Porous Media of Goaf

PONE-D-24-39974R2

Dear Dr. zhuo,

We’re pleased to inform you that your manuscript has been judged scientifically suitable for publication and will be formally accepted for publication once it meets all outstanding technical requirements.

Kind regards,

Marko Čanađija

Academic Editor

PLOS ONE
---

## [Editor Report · Acceptance letter]

8 Dec 2024

PONE-D-24-39974R2 

PLOS ONE

Dear Dr. Zhuo, 

I'm pleased to inform you that your manuscript has been deemed suitable for publication in PLOS ONE. Congratulations! Your manuscript is now being handed over to our production team.

Kind regards, 

on behalf of

Dr. Marko Čanađija 

Academic Editor

PLOS ONE